# The MAPK/ERK channel capacity exceeds 6 bit/hour

**Paweł Nałęcz-Jawecki**[1], **Paolo Armando Gagliardi**[2], **Marek Kochańczyk**[1], **Coralie Dessauges**[2¤], **Olivier Pertz**[2]*, **Tomasz Lipniacki**[1]*

**1** Institute of Fundamental Technological Research, Polish Academy of Sciences, Warsaw, Poland,
**2** Institute of Cell Biology, University of Bern, Bern, Switzerland

¤ Current address: Laboratory of Systems Pharmacology, Harvard Medical School, Boston, Massachusetts, United States of America
* olivier.pertz@izb.unibe.ch (OP); tlipnia@ippt.pan.pl (TL)

## Abstract

Living cells utilize signaling pathways to sense, transduce, and process information. As the extracellular stimulation often has rich temporal characteristics which may govern dynamic cellular responses, it is important to quantify the rate of information flow through the signaling pathways. In this study, we used an epithelial cell line expressing a light-activatable FGF receptor and an ERK activity reporter to assess the ability of the MAPK/ERK pathway to transduce signal encoded in a sequence of pulses. By stimulating the cells with random light pulse trains, we demonstrated that the MAPK/ERK channel capacity is at least 6 bits per hour. The input reconstruction algorithm detects the light pulses with 1-min accuracy 5 min after their occurrence. The high information transmission rate may enable the pathway to coordinate multiple processes including cell movement and respond to rapidly varying stimuli such as chemoattracting gradients created by other cells.

## Author summary

To respond to external stimulation, living cells need to transfer information from membrane receptors to downstream proteins that regulate physiological outcomes. In this study we used cells expressing a light-activatable receptor that transmits signals to MAPK/ERK transduction pathway, which regulates numerous physiological processes including proliferation, differentiation and cell motility. By stimulating these cells with various sequences of short light pulses we found that ERK, the last component of the considered signal transduction pathway, may receive at least 6 bits per hour. This bitrate allows 6 binary decisions per hour, enabling rapid responses to varied extracellular stimuli.

## Introduction

Living cells communicate with each other and constantly monitor their extracellular milieu. Upon receiving a stimulus, they have to recognize its identity and act accordingly to operate in a coordinated manner and properly adapt to the changing environment. At the molecular

---

**Data Availability Statement:** A complete set of live-cell imaging data is available at DOI: 10.5281/zenodo.7808385 Source code used to generate main figures is available at URL: https://github.com/pawelnalecz/pulsatile-information.

**Funding:** This study was supported by Swiss Cancer League (https://gap.swisscancer.ch) grant KLS-4867-08-2019 to OP, Swiss National Science Foundation (https://www.snf.ch) grant Div3 310030_185376 to OP, and National Science Centre Poland (https://www.ncn.gov.pl) grant 2019/35/B/NZ2/03898 to TL. The funders had no role in study design, data collection and analysis, decision to publish, or preparation of the manuscript.

**Competing interests:** The authors have declared that no competing interests exist.

level, recognition of a specific signal as well as its reliable transmission and appropriate interpretation involve diverse intracellular processes that are notoriously stochastic [1]. At the single-cell level, major intracellular signaling pathways respond to an external stimulus in a crude all-or-nothing manner [2–5]. The inability of individual cells to resolve the level of stimulation is reflected in the amount of information about the ligand concentration the signaling pathways can transmit, which has been estimated to only slightly exceed 1 bit [6–9].

However, when cells are subjected to temporally varying stimulation by morphogens, hormones, or cytokines, it is of interest to quantify the maximum rate at which information can be transferred through a biochemical signaling pathway. A similar question has been asked before in the fields of neuroscience and bacteria or amoebae chemotaxis. Neurons utilize precise timing of spikes to evoke proper response [10–12]. The information transmission rate (or simply bitrate) through a single neuron may reach tens of bits per second [13,14]. For bacteria navigating in a chemical gradient, the drift velocity scales linearly with the gradient, while the bitrate was found to be proportional to the square of the gradient [15]. When concentration varies on the length scales of millimeters to centimeters, the chemotactic system of *E. coli* transfers about 0.01 bit per second, which allows bacteria to climb up the gradient with a few percent of its swimming speed. This suggests that at low gradients, drift velocity is information limited [15]. Without explicitly quantifying the information flow, Meier *et al.* [16] showed that *Dictyostelium discoideum* amoebae may synchronously readapt their migration direction to switching chemotactic gradient oscillating at rates up to 0.02 Hz. This suggests that the rate of information transmission to the navigation system is up to 1 bit per minute, and the limit is achieved if all cell decisions are correct. The maximum bitrate depends on biophysics of a specific information channel. In the case of neurons, where the bitrates are highest, information is transmitted with the help of fast electro-chemical signals; in the case of *E. coli* and MAPK, information is transmitted by means of posttranslational proteins modifications, thus expected bitrates are lower. The bitrate observed in the *E. coli* chemotactic system can be higher than that expected for the MAPK channel, because bacteria are much smaller than a eukaryotic cell. Some other signaling pathways such, as that of NF-κB or p53, require protein synthesis and degradation, which are slower than post-translational modifications, and so for these pathways even lower bitrates are expected.

Gagliardi *et al.* [17] demonstrated that the ERK pulses triggered locally by growth factors shed by apoptotic cells coordinate the maintenance of epithelial integrity. *In vivo* studies showed pulsatile ERK dynamics during wound healing in mice, wherein the wound edge initiates recurrent, quasiperiodic mechanochemical traveling fronts of ERK activity [18]. Such waves were also observed during mammalian epidermis development [19] and during scale regeneration in zebrafish [20]. Pulsatile ERK dynamics has been observed also for tonic EGF stimulation, for example Albeck *et al.* [21] and Aoki *et al.* [22] have shown that the MAPK pathway converts various types of stimulation into ERK pulses, suggesting their role in generating physiological outcomes.

In this study, we assessed the capability of the MAPK/ERK pathway to transduce information encoded in a sequence of stimulatory pulses, and estimated the lower bound for the rate of information transmission through this pathway. In light of the mentioned findings, we focused solely on pulsatile stimulation protocols. As we conjecture that protocols ensuring highest information transfer are used for cell signaling, we searched for a protocol that would maximize the information transmission rate, and thus estimated the MAPK/ERK channel capacity. We considered three different families of pulsatile input protocols: (i) *binary encoding*, resembling digital communication with zeros and ones occurring in well-defined, temporally equidistant time slots, is plausibly the simplest pulsatile stimulation protocol; (ii) *interval encoding*, with exponentially-distributed waiting times in discrete time domain (so, effectively,

geometrically-distributed waiting times), is statistically maximally random (entropic); this protocol is also natural form the biological perspective as it implies constant propensity of receiving stimulation pulses in time; (iii) *interval encoding with an additional gap* after each pulse is an attempt to tailor a maximally entropic stimulation to cells characterized by an inextricable refractory (resting) period. We used an epithelial cell line with an optogenetically modified fibroblast growth factor receptor (optoFGFR) [23,24] which allowed us to stimulate the pathway with short light pulses activating ERK with the similar kinetics as EGF or FGF [25]. The cell line is stably transfected with a reporter, enabling fine temporal monitoring of ERK activity. By stimulating cells with random pulse trains, generated with probabilistic algorithms according to protocols from the three families, and applying classifier-based algorithm to reconstruct the stimulatory sequence, we demonstrated that the maximum information transmission rate through the pathway exceeds 6 bits per hour. A classifier-free approach gives an even higher estimate of about 8.5 bits per hour.

## Results

### Information encoding and information transmission rate

The maximum amount of information that can be transmitted through a communication channel per unit time using the best possible input protocol is called the channel capacity $C$:

$$C = \max_{\text{protocols}} \lim_{\Delta t \to \infty} \frac{I(S, R)}{\Delta t} \tag{1}$$

where $I(S, R)$ is the mutual information between the input (signal) $S$ and the output (response) $R$ in an experiment of duration $\Delta t$ [26]. In this work, we computed $I(S, R)$ as the difference:

$$I(S, R) = H(S) - H(S|R) \tag{2}$$

where $H(S)$ is the input entropy (the amount of information sent), which in our experimental design can be computed directly from the assumed probabilistic distribution of the pathway-stimulating input sequence; and $H(S|R)$ is the conditional entropy (the amount of information lost), which captures the uncertainty introduced by the channel (here: the signaling pathway). The value $H(S|R)$ depends both on the channel and the signal and as such is estimated based on experimental data.

There are multiple ways to encode information in a train of pulses. When the magnitudes of all stimulation pulses are equal, the input can be perceived as a sequence of 0s (no pulse) and 1s (pulse). The input information (or entropy) rate increases with the frequency of pulses, reaching the maximum for binary sequences that have equal probabilities of 0 and 1. However, shorter intervals between pulses imply higher information loss due to imperfect transmission. One should thus expect that there exists an optimum, for which the bitrate is maximal.

We applied three types of pulsatile stimulation protocols to estimate the lower bound for the channel capacity of the MAPK/ERK pathway (see Fig 1A and Materials and methods for details). In what we call the *binary encoding protocol*, information is encoded in a sequence of 0s and 1s sent at regular time intervals $\tau_{\text{clock}}$ with equal probabilities. In our experiments, we stimulated cells with light pulses according to a 0/1 sequence of length 19: 1011010011110000101 and its logic negation (both of them containing each of the 16 possible subsequences of length 4 exactly once, which allowed us to collect good statistics in a single, relatively short experiment). In the *interval encoding protocol*, information is encoded in the lengths of intervals between subsequent input light pulses. To maximize the input entropy (which, in light of Eq (2), bounds the transmitted information from above), the intervals were drawn from geometric distribution. In the *interval encoding protocol with a minimal gap*, the

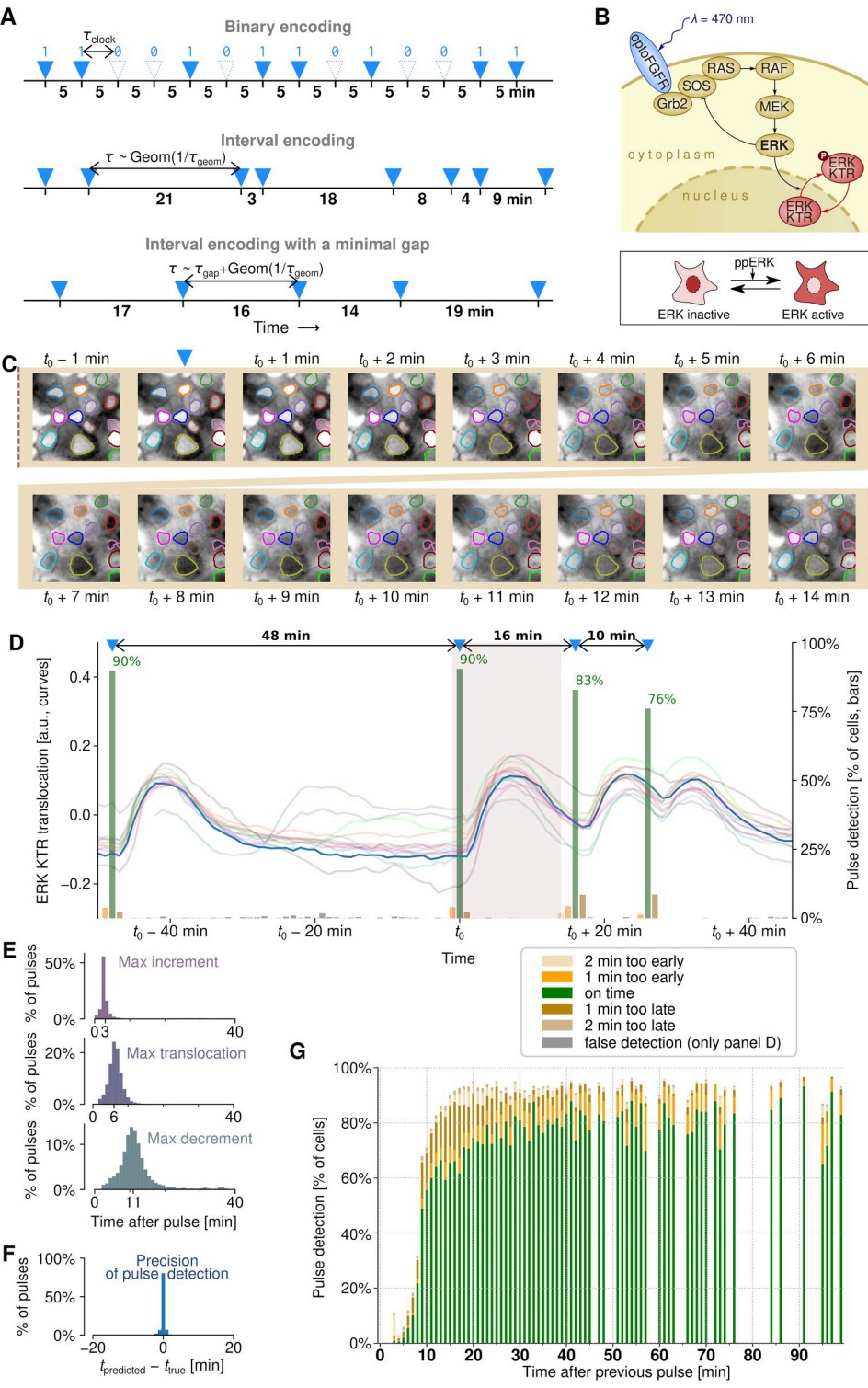

**Fig 1. Decoding pulsatile signals relayed through the MAPK/ERK pathway. (A)** Input: three considered protocols of encoding information in a pulse train. In binary encoding, a pulse is either present or not in each of the temporally equidistant slots. In interval encoding, information is carried by the lengths of time intervals between subsequent pulses. The intervals are drawn from a geometric distribution (with one-minute resolution). In interval encoding with a minimal gap, the intervals are drawn from a geometric distribution and a fixed "gap" is added. Small numbers in bold typeface denote intervals between pulse slots (in binary encoding) and inter-pulse intervals (in both interval encodings). **(B)** Diagram of the MAPK/ERK pathway within the engineered MCF-10A cells. Blue light activates

optoFGFR, triggering a kinase cascade, which culminates in ERK activation. The fluorescent (mRuby2) ERK KTR, which in non-stimulated cells is mostly localized to the nucleus, upon phosphorylation by active nuclear ERK (ppERK) is exported to the cytoplasm. (A drop of) the mean nuclear fluorescence of ERK KTR is used as a proxy of its translocation and ERK activity. **(C)** Output: ERK KTR translocation in response to activation of optoFGFR by a light pulse at $t_0$. Nuclear contours of 13 cells are marked with different colors. For the sake of presentation, microscopic images are normalized such that the original 10%–90% quantiles of pixel intensity span over the whole grayscale range, black to white. **(D)** Input reconstruction: time track of ERK KTR translocation in response to a sequence of 4 light pulses in the 13 cells marked in panel C. The shaded interval is the interval in which the snapshots shown in panel C were acquired; the trajectories correspond to respective color-coded nuclear outlines shown in panel C. Green bars show the proportion of cells (estimated based on 400 single cell trajectories) in which a pulse was detected by the trained classifier. **(E)** Histograms of three basic temporal features characterizing the ERK KTR translocation profile: time to the largest translocation increment, the peak of translocation, and the largest translocation decrement (all with respect to the time of a light pulse that elicited the characterized response). In a typical cell, the translocation has the steepest slope between 2 and 3 min after the light pulse, reaches the maximum at 6 min, and rebounds at the highest rate between 10 and 11 min after the light pulse. Data from all 6 experiments with the interval encoding protocol. **(F)** Accuracy of the light pulse detection. Most of the pulses are detected exactly with one-minute resolution. Data as in panel E, classifier trained on other cells from the same experiment. **(G)** Proportion of cells in which a pulse was detected as a function of the interval after the previous pulse. Data as in panel F.

time intervals were also drawn from geometric distribution (with the average inter-pulse interval $\tau_{geom}$), and then $\tau_{gap}$ was added to the drawn value, thus the expected value of the inter-pulse interval was $\tau_{geom} + \tau_{gap}$. For each $\tau_{gap}$, we chose $\tau_{geom}$ that maximizes the input entropy (see Materials and methods for details). Introduction of $\tau_{gap}$ reduces the input entropy rate compared to the interval encoding protocols, but increases the percentage of cells evoking a timely response to a light pulse, which facilitates accurate information transmission.

## Detection of pulses

To quantify the bitrate through the MAPK/ERK pathway, we used the human mammary epithelial cell line MCF-10A with both an optogenetically modified fibroblast growth factor receptor (optoFGFR) [17,25], which can be activated with blue light (470 nm), and a fluorescent ERK kinase translocation reporter (ERK KTR), which translocates from the nucleus to the cytoplasm when phosphorylated by ERK (Fig 1B). The cells were stimulated with short (100 ms, see Materials and methods) blue light pulses according to the three types of encoding protocols (Fig 1A). ERK KTR translocation was measured at one-minute resolution (Fig 1C and 1D; see Materials and methods and S1 Fig for details of the workflow). We observed that the signal was transduced from optoFGFR to ERK KTR within single minutes, consistently with Ref. [25], with the steepest translocation increment (drop of the nuclear ERK KTR fluorescence) observed between 2 and 3 min, maximum translocation at 6 min, and steepest translocation decrement around 11 min after the light pulse (Fig 1E).

We then employed a classifier based on the *k*-nearest neighbors (*k*NN) algorithm (see Materials and methods for details) to reconstruct the sequence of stimulatory light pulses using the quantified single-cell trajectories of nuclear ERK KTR (Fig 1D). The algorithm was designed to predict the timing of light pulses using a short (overall, 8 minute-long) window sliding over the ERK KTR translocation trajectory. The method was typically able to predict the pulse timing with one-minute resolution (Fig 1F), markedly better than any individual pulse feature (*cf*. Fig 1E and 1F). The percentage of detected pulses increases sharply with the time interval between the pulse to be detected and the previous pulse, reaching about 70% for the interval of 10 min and nearly 90% for the interval of 15 min (Fig 1G). Missed pulses, pulses detected as occurring earlier or later (proportion of which is shown in Fig 1G), and false detections are responsible for the information loss.

## Bitrate estimation

We computed the information loss along the pathway as the entropy of the input conditioned on its (classifier-based) reconstruction from single-cell ERK KTR trajectories, $H(S|R)$. Next, by subtracting the information loss from the input entropy $H(S)$, we quantified the transmitted information $I(S|R)$, and eventually, by dividing the result by experiment duration, we estimated the bitrate. It is important to stress that our method of pulse detection is based only on an 8-min rolling window, hence we do not use prior knowledge about the minimal gap between pulses.

In Fig 2A we show 10 example single-cell ERK KTR translocation trajectories of cells responding to the first sequence used in the binary encoding protocol with the time span between subsequent digits (clock period) $\tau_{clock}$ = 10 min. The percentage of cells identified as responding to light pulses varied from around 80% to nearly 100%. As expected, the detectability is higher when the time span from the preceding pulse is 20 min or more (more than 95% pulses are detected). A tiny percentage of cells, $\leq$ 1%, was incorrectly identified as responding at time points without a light pulse.

For the binary protocol we performed experiments for 7 different clock periods ranging from 3 to 30 min (Fig 2B). The input information rate, 1 bit/$\tau_{clock}$, decreases with increasing $\tau_{clock}$ but, as expected, for longer $\tau_{clock}$ the fraction of information lost due to missed pulses or false detections is lower. For the shortest $\tau_{clock}$ of 3 min, the input information rate reached 20 bit/h, but due to severe information loss the transmitted information rate was lower than for $\tau_{clock}$ = 5 min (Fig 2B). It should be noted that in the case of the binary encoding protocol during input reconstruction we utilize the *a priori* knowledge of the time points in which the light pulses could have occurred ("pulse slots"). Thus, based on the ERK KTR translocation trajectory we have to decide only whether the light pulses occurred in these time points or not. This makes the reconstruction easier than in the interval encoding protocols, however binary encoding appears rather artificial in the biological context.

In Fig 2C we show a representative part of ERK KTR translocation trajectories in 10 cells responding to stimulation with light according to the interval encoding protocol with $\tau_{geom}$ = 32 min. For all but one input pulse the percentage of cells identified as responding timely exceeds 75% and is generally higher for pulses occurring after a longer time span from the previous pulse. For inter-pulse intervals below 10 min, the detectability decreases rapidly (for example, the pulse at minute 744 that occurred 8 min after its predecessor was detected timely by only 25% of cells). For each pulse there is also a fraction of detections that indicate an input pulse 1 min before or after the true pulse.

We performed 6 experiments according to the interval encoding protocol with $\tau_{geom}$ ranging from 22 to 55 min (Fig 2D). The highest bitrate of 6.3 bit/h was found for $\tau_{geom}$ 36 min, while for $\tau_{geom}$ = 32 and 42 min the bitrate was found equal 5.3 and 5.0 bit/h, respectively. The shortest (22 min) and longest (55 min) average intervals resulted in, respectively, the highest and the lowest information loss. This causes that for these two suboptimal protocols, having respectively high (15 bit/h) and low (7.7 bit/h) input information rate, the bitrate is similar, between 4 and 5 bit/h. The information is lost due to the imprecise detections (that carry incomplete information about the timing of pulses), missed pulses and false detections (Fig 2D). The estimation of the information loss due to these three types of signal reconstruction errors is outlined in S2 Fig, panels C–E, and described in Materials and methods.

In Fig 2E, we show 10 example trajectories for the interval encoding protocol with $\tau_{geom}$ = 10 min and a gap of $\tau_{gap}$ = 20 min (giving together the mean inter-pulse interval equal 30 min). The ERK KTR translocation pulses are well distinguishable and thus in around 90% of single-cell trajectories the stimulation pulses were reconstructed with one-minute resolution.

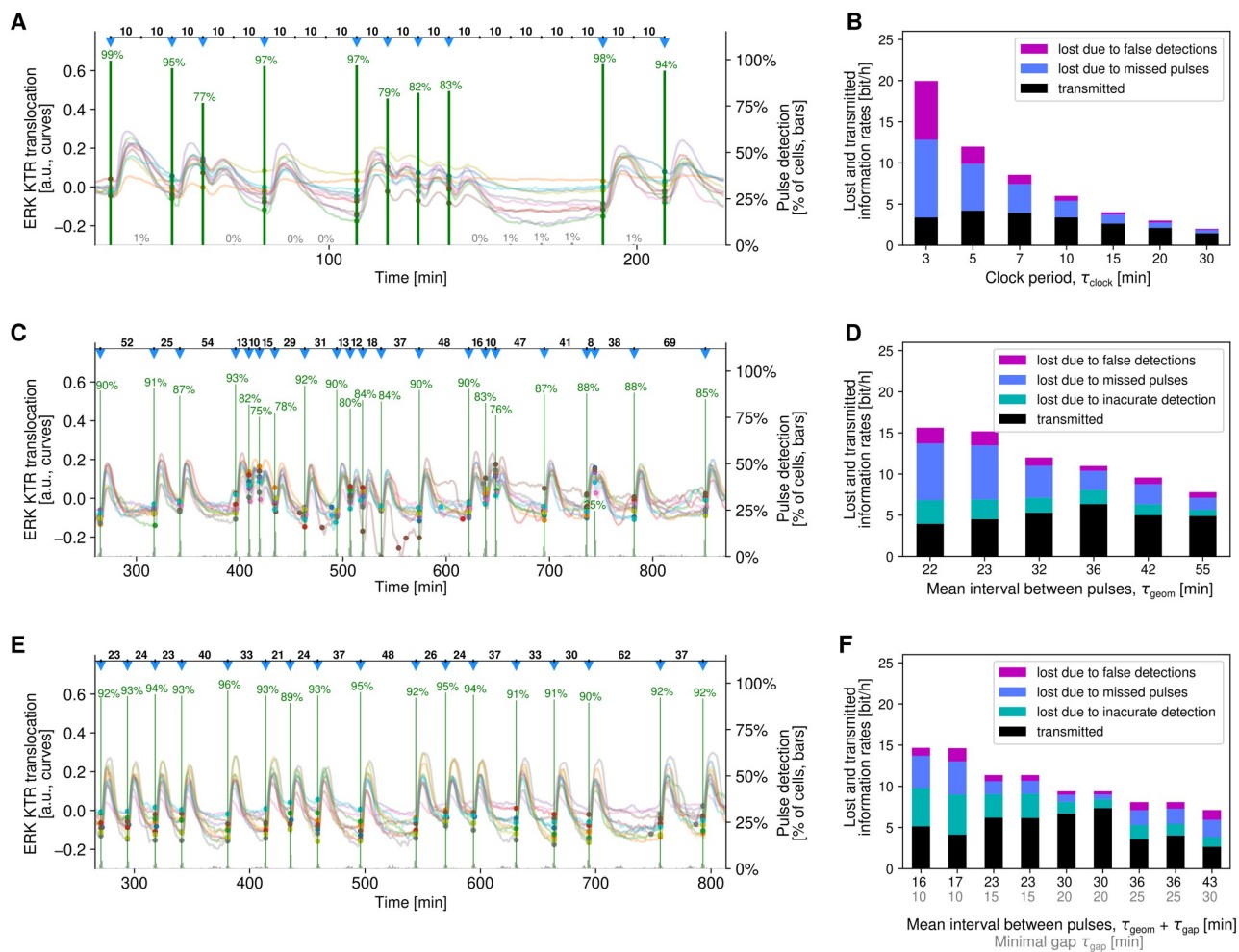

**Fig 2. Pulse detection and bitrate estimation in three protocols. (A)** Binary encoding: 10 representative trajectories from an experiment with slots every $\tau_{clock}$ = 10 min. Stimulation pulses are marked with blue down-pointing triangles. Small filled circles indicate light pulses detected based on the corresponding trajectories. Percentage-labeled green bars in each pulse slot show the fraction of cells in which a pulse was correctly detected. Fractions of false positive detections are shown as percentage-labeled gray bars. **(B)** Binary encoding: bitrate and sources of bitrate loss in experiments with various clock periods. In each column, total height of the stacked bars is equal to the input entropy rate. The number of analyzed cell trajectories was 3200 for clock periods $\tau_{clock}$ ranging from 3 to 15 min and 1200 for $\tau_{clock}$ = 20 and 30 min. Each analyzed cell trajectory contained 19 pulse slots. **(C)** Interval encoding: 10 representative trajectories from the experiment with the mean interval $\tau_{geom}$ = 36 min. Stimulation pulses are marked with blue down-pointing triangles. Colored dots indicate light pulses detected based on the corresponding trajectories. Green bars show the fraction of cells in which a pulse was detected in a particular minute. **(D)** Interval encoding: bitrate and sources of bitrate loss in experiments with various mean intervals. In each column, total height of the stacked bars is equal to the input entropy rate. The number of analyzed cell trajectories was 400 in each of 6 experiments. The mean number of light pulses in analyzed trajectories was in the range 24–60 depending on the mean interval (all experiments lasted 27–33 h). **(E)** Interval encoding with a minimal gap: 10 representative trajectories from the experiment with a minimal gap of $\tau_{gap}$ = 20 min and the inverse of the optimized geometric distribution parameter $\tau_{geom}$ = 10 min. Graphical convention as in panel C. **(F)** Interval encoding with a minimal gap: bitrate and loss sources in experiments with various minimal gaps and mean intervals. In each column, total height of the stacked bars is equal to the input entropy rate. The number of analyzed cell trajectories was 400 in each of 9 experiments. The mean number of light pulses in analyzed trajectories was in the range 19–96 depending on the mean interval (all experiments last 27–33 h).

For this protocol the bitrate was found to be the highest, equal 7.0 bit/h based on two experimental replicates (Fig 2F). The somewhat lower bitrate of 6.2 bit/h was found for the protocol with $\tau_{geom}$ = 8 min and $\tau_{gap}$ = 15 min (average of two replicates). As for the previous protocol, information is lost due to the three types of errors.

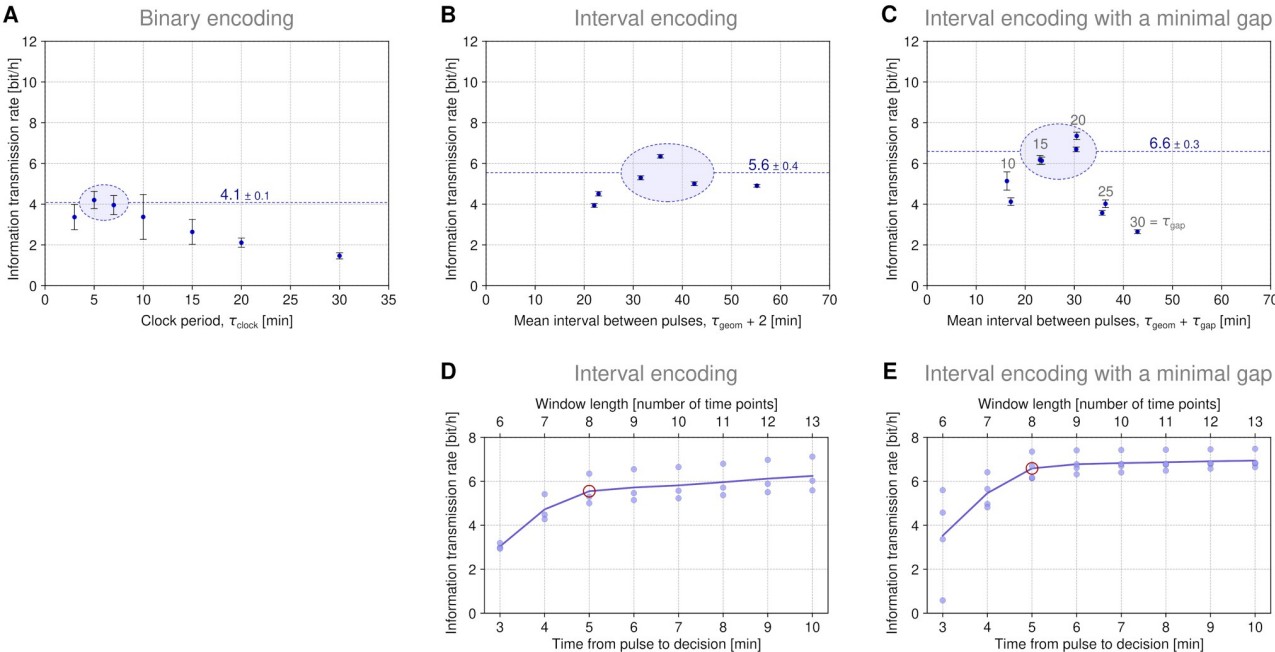

**Fig 3. Channel capacity estimation–reconstruction-based approach. (A)–(C)** Information transmission rate for the three protocol families. In A, each data point is a mean of 8–10 experiments, and standard deviation of these replicates is shown. In B and C, where each point corresponds to a single replicate, the standard deviation of the estimates obtained after 10 random train–test splits of the 400 cell trajectories is shown. The maximum bitrate for a given protocol is estimated (from below) as the mean ± standard error of the mean of 2–4 encircled points and marked with a dashed line. **(D)–(E)** Lower bound of maximum bitrate as a function of time from pulse to its detection for two interval encoding protocols. Points correspond to experiments encircled in panels B and C; line shows the average over these experiments. Pulse detection is based on the ERK KTR trajectory window that begins 2 min before a pulse slot and ends at the time after the pulse slot indicated on the horizontal axis (classification is always based on three overlapping slices of their length adjusted to the length of the window). The 8 min-long window (red circle) was used for pulse detection throughout the paper.

## Latency and accuracy of signal detection

In Fig 3A–3C we gathered all the experiments to estimate the maximum bitrate that can be achieved across all 3 protocols. From 6 to 9 experiments performed for each protocol we selected 2–4 experiments for which we obtained the highest bitrate, and estimated the lower bound of the maximum bitrate as the average over these selected experiments. As a result, we obtained the following bounds (mean ± standard error of the mean):

- 4.1 ± 0.1 bit/h for the binary encoding protocol with the optimal clock period ($\tau_{clock}$) in the range of 5–7 min,

- 5.6 ± 0.4 bit/h for the interval encoding protocol with the optimal average interval ($\tau_{geom}$) in the range of 32–42 min, and

- 6.6 ± 0.3 bit/h for the interval encoding protocol with a minimal gap, with the optimal gap interval ($\tau_{gap}$) in the range of 15–20 min and the (inverse of the) corresponding geometric distribution parameter ($\tau_{geom}$) in the range of 8–10 min.

Altogether, the above results show that the channel capacity of the MAPK/ERK pathway exceeds 6 bit/h and suggest that the highest bitrate can be achieved for the interval encoding protocol with a gap.

We should mention that the bitrate estimations were obtained after rejecting the fraction $f$ = 20% of ERK KTR trajectories that exhibit the lowest variability, that is, those with the smallest average square change of the nuclear intensity between subsequent snapshots. As we can see in S3 Fig, panel A, the maximum bitrate estimated for each of three protocols is an increasing function of $f$ for $f$ in the range from 0 to 50%. In the case of the binary encoding, the maximum bitrate increases nearly linearly with $f$, which results from the fact that for short clock periods lower variability directly corresponds to lower chance of responding to a light pulse. However, for the interval encoding protocol the most significant increase is observed in the range 0–20%, which motivated us to remove 20% trajectories. At least some of these trajectories come from cells with optoFGFR or ERK KTR not functioning properly.

In S3 Fig, panels B–D, we show the histograms of bitrate measured for single cells in all experiments for the three encoding protocols. The orange area in the histograms corresponds to the removed 20% fraction of cells. We can notice that the histograms for the interval encoding protocol with a minimal gap are bimodal, but by removing the 20% of cells we may eradicate the lower mode, which additionally justifies the choice of the particular value of $f$ (for consistency we removed the same fraction of trajectories from experiments for the other two protocols).

All bitrate estimations are based on the ERK KTR translocation trajectory analyzed in an 8 min-wide rolling window that ends $t_D$ = 5 min after the hypothetical pulse to be detected ($t_D$ stands for 'time to decision'). This means that we account for information that is available for each cell 5 min after the pulse that the cell is expected to recognize. In Fig 3D we give the maximal bitrate estimates (for the interval encoding protocol with a minimal gap jointly for the experiments with the highest bitrate, encircled in Fig 3A–3C) as a function of $t_D$. We found that increasing $t_D$ beyond 5 min only marginally increases the bitrate, while decreasing it below 3 min dramatically decreases the bitrate (which possibly reflects the fact that the maximum increment is usually observed between 2 and 3 min after the pulse).

## Reconstruction-free bitrate estimation

An advantage of the reconstruction-based approach is that it enables discerning particular sources of classification errors (as shown in Fig 2B, 2D and 2F) and provides insight into information available to a responding cell. On the other hand, any classifier loses some information about the responding cell and thus provides a lower bound on the estimate of the cell capability of transmitting information. Therefore (prompted by an anonymous Reviewer), we performed a direct (reconstruction-free) estimation of the information transmission rate for both families of interval encoding protocols. We used the $k$NN ($k$ = 20) algorithm to estimate the conditional entropy $H(S|R)$, where $R$ is the quantified response, and obtain $I(S,R) = H(S) − H(S|R)$. Then, by dividing the result by experiment duration, we estimated the bitrate, see Materials and methods. As in the reconstruction-based approach, we used an 8-min rolling window. As shown in Fig 4A and 4B, the reconstruction-free estimates are about 2 bit/h higher than those obtained in the reconstruction-based approach, and equal (mean ± standard error of the mean):

- 8.2 ± 0.1 bit/h for the interval encoding protocol (based on four data points encircled in Fig 4A) and

- 8.5 ± 0.2 bit/h for the interval encoding protocol with a minimal gap (based on six data points encircled in Fig 4B).

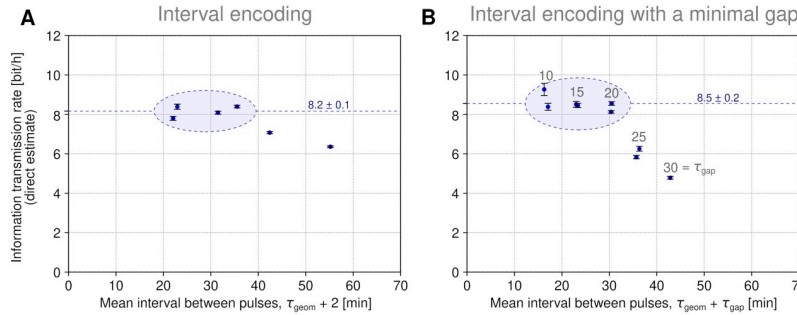

**Fig 4. Channel capacity estimation–reconstruction-free approach. (A)–(B)** Information transmission rate for two families of interval encoding protocols. Error bars indicate the standard deviation of the estimates obtained after 10 random resamplings of 50% (200 out of 400) cell trajectories. The maximum bitrate for a given protocol is estimated as the mean ± standard error of the mean of 4 or 6 encircled points and marked with a dashed line.

The largest difference between the reconstruction-based and the reconstruction-free approach can be observed for data points corresponding to the shortest mean intervals (*cf.* Figs 3B and 4A). Heuristically, it can be explained as follows: In the reconstruction-based approach, the small missed pulse errors (occurring for pulses preceded by long intervals) are averaged with large missed pulse errors (occurring for pulses preceded by short intervals). Thus, for short mean intervals, the average fidelity of the pulse reconstruction is relatively low which implies high $H(S|R)$ calculated from the confusion matrix. To illustrate this with a simple example: when one sends 20 zero-one digits, and the first 10 of them are received with $p = 1$ confidence, while remaining with $p = \frac{1}{2}$ confidence, then the transmitted information is 10 bits (provided that the receiver knows which bits should be trusted). However, when the receiver does not know which bits should be trusted, the averaged confidence is $p_a = \frac{3}{4}$, which implies that the transmitted information is $20 (1 - H(S|R)) = 20 (1 + p_a \log_2(p_a) + (1 - p_a) \log_2(1 - p_a)) \approx 3.8$ bit. In the reconstruction-free approach the uncertainty about pulses is not averaged.

## Discussion

Toettcher *et al.* [27] showed that the RAS–ERK module of the MAPK pathway may transmit input signals across a broad range of timescales, from 4 min to multiple hours. Here, by applying signal reconstruction-based approach we estimated the information transmission rate from FGFR to an ERK activity reporter to find that the MAPK/ERK channel capacity exceeds 6 bit/h. In a direct, reconstruction-free approach, we obtained a higher estimate of 8.5 bit/h, which shows that at the signal reconstruction step a substantial portion of information is lost. It is debatable, however, whether cells can utilize the input information without "reconstructing the signal".

We analyzed three protocols to encode information: binary encoding, interval encoding, and interval encoding with a minimal gap, to find that interval-encoded information can be transmitted at a significantly higher bitrate than binary-encoded information. The presence of a minimal gap between pulses of length comparable to the refractory time (*i.e.*, time in which the cell may not unambiguously respond to a subsequent pulse) further increases the bitrate (although this increase is small for the reconstruction-free approach). The lower bound of the MAPK/ERK channel capacity found in this study is well above our earlier purely theoretical estimates [28], which suggested a bitrate between 1.4 and 1.9 bit/h. Those MAPK-model estimates were obtained for the binary protocol, but the main source of discrepancy is the longer

refractory time of ~50 min, arising in the computational model [29]. Interval encoding appears the most natural from the biological perspective as it assumes that pulses occur independently with some constant intensity (the mean interval is the inverse of the intensity). These random pulses can be associated with single-cell apoptosis, which recently has been found to trigger synchronous ERK activation [30] that propagates radially for about three cell "layers" [17].

Using the reconstruction-based approach we found that for the interval encoding with a minimal gap the bitrate may exceed 6 bit/h, which can be considered as a "safe" lower bound on the MAPK/ERK channel capacity. The fact that interval encoding is associated with the highest bitrate follows from the cells' ability to respond to light pulses in a very synchronous manner. The light pulses can be detected nearly perfectly with one-minute resolution (provided that the time from the preceding pulse is longer than the refractory time), based on the ERK activity in an 8-min interval ending 5 min after the light pulse. Synchronous responses to the stimuli may enable long-distance waves of ERK activity that assist wound healing, inducing collective cell migration [18,31]. In our experiments we do not observe communication between cells, because all cells are illuminated equally and simultaneously. However, when cells are stimulated in a single spot only, for example due to an apoptotic event, one may observe propagation of an ERK activity wave outside of this spot [17,32]. This clearly indicates for cell-to-cell communication and propagation of information. One could expect that even if such wounding- or apoptosis-generated waves are initiated at small time intervals, the first "layer" of cells filters out the pulses that occur within the refractory time, and thus the farther "layers" of cells will be subjected to stimulation that resembles the interval encoding with a minimal gap.

Since we have not investigated signaling with resolution finer than 1 min we may not rule out that the MAPK/ERK channel capacity exceeds the determined lower bound of 6 bit/h. Using finer temporal discretization would increase the entropy of the input, but in the reconstruction-based approach it would also result in more classification errors, therefore both $H(S)$ and $H(S|R)$ would be increased. Even for one-minute resolution, a substantial part of information loss is due to inaccurate detections (see Fig 2D and 2F).

In our analysis we lumped together responses of different cells. This is known to reduce the measured information transfer because cell-to-cell differences in responses are treated as noise. As observed earlier by Toettcher *et al.* [27], individual cells exhibit precise and reversible Ras/ERK dose-response behavior; in our case we observe that there are faster and slower cells, and this causes that we occasionally reconstruct pulses with 1 minute delay or 1 minute in advance. By learning about each individual cell from its response to first light pulses, one could use its response to reconstruct subsequent pulses with higher accuracy; such type of analysis was performed by Selimkhanov *et al.* [7]. However, when one aims at interpreting transmitted information as the ability of cells to exhibit synchronous responses to stimuli (required for ERK activity wave propagation), cell-to-cell differences can be interpreted as an (extrinsic) noise.

Cells send, receive, and process information, however a natural question is how the information transmitted at the rate exceeding 6 bit/h can be utilized. Intuitively, high information rate is critical when a cell has to respond to constantly changing stimuli. Neurons were shown to transmit information at rates of tens of bits per second [13,14]. *E. coli* chemotactic system transfers about 0.01 bit per second, and this information rate limits its drift velocity in shallow chemical gradients [15]. Meier *et al.* [16] showed that *D. discoideum* (chemotaxis of which is known to be regulated by an atypical MAP kinase, ERKB [33,34]) may navigate in a switching chemotactic gradient, changing its direction within a minute. This suggests the information processing rate of about 1 bit per minute, or less if some cell switch-or-not decisions are incorrect.

In higher organisms, the MAPK/ERK pathway is known to coordinate diverse complex behaviors and cell fate decisions, such as cell proliferation, differentiation, migration, senescence and apoptosis [35]. The distinction between particular cell fates depends on ERK pulsatile activity integrated over timescales of multiple hours [36]. RAS-to-ERK cascade transmits signals (and information) from numerous inputs to numerous outputs regulated by ERK and its upstream components, MEK, RAF, and RAS. During collective cell migration, ERK controls myosin contractile pulses [37]. Spatially localized RAS activation is responsible for formation of actin-based cell protrusions [38], which arise and contract at the timescale of single minutes [39], directing cell motion. Since ERK works as a RAS-activated (via RAF and MEK) RAS inhibitor [40], the ERK activity profile influences cell protrusion and retraction, and thus cell motion [41]. The high information capacity of the MAPK/ERK channel demonstrated in this study enables coordination of the versatile functions associated with ERK and its upstream kinases, with cell motion in response to rapidly changing stimuli likely being the process the most information is fed into.

## Materials and methods

### Cell lines

The MCF-10A human mammary epithelial cells (derived from an adult female) were a gift of J.S. Brugge. The cells were modified to stably express the nuclear marker H2B-miRFP703, the ERK biosensor ERK-KTR-mRuby2 [42] and the optogenetic actuator optoFGFR [23] and used before in Refs. [17,25].

### Cell culture

The modified MCF-10A cells were cultured in tissue-culture treated plastic flasks and fed with a growth medium composed of DMEM:F12 (1:1) supplemented with horse serum 5%, recombinant human EGF (20 ng/ml, Peprotech), L-glutamine, hydrocortisone (0.5 µg/ml, Sigma-Aldrich/Merck), insulin (10 µg/ml, Sigma-Aldrich/Merck), penicillin (200 U/ml) and streptomycin (200 µg/ml). The cells were routinely split upon reaching ~90% confluency. For time-lapse optogenetic experiments, cells were seeded on 24-well 1.5 glass bottom plates (Cellvis) coated with 5 µg/ml fibronectin (PanReac AppliChem) at $1 \times 10^5$ cells/well density in growth medium two days before the experiment. Four hours before the optogenetic stimulation (2.5 h before imaging), cells were washed twice with PBS and then cultured in a starvation medium composed of DMEM:F12 (1:1) supplemented with BSA (0.3% Sigma-Aldrich/Merck), L-glutamine, hydrocortisone (0.5 µg/ml, Sigma-Aldrich/Merck).

### Optogenetic stimulation

OptoFGFR was stimulated with pulses of blue LED light (100 ms, 470 nm, 3 W/cm$^2$) applied according to specific stimulation protocols (see a further subsection on stimulation protocols). It should be noticed that for short pulses the response depends on the total pulse energy per unit surface, and that 0.3 J/cm$^2$ is above the saturation dose [25].

### Imaging

Imaging experiments were performed on an epifluorescence Eclipse Ti inverted fluorescence microscope (Nikon) controlled by NIS-Elements (Nikon) with a Plan Apo air 20× (NA 0.8) objective. Laser-based autofocus was used throughout the experiments. Image acquisition was performed with an Andor Zyla 4.2 plus camera at the 16-bit depth every 1 min. The following excitation and emission filters were used: far red: 640 nm, ET705/72m; red: 555 nm, ET652/

60m; green: 470 nm, ET525/36m. Imaging started 1.5 h before the onset of optogenetic stimulation (to provide proper history for ERK KTR track normalization, see a further subsection on signal processing).

## Stimulation protocols

In the case of binary encoding, all stimulation protocols have 19 temporally equidistant pulse "slots". We applied the following fixed pattern of pulses: 1011010011110000101 (1 = pulse, 0 = no pulse), which is the shortest sequence containing uniquely every possible subsequence of four binary digits. For each "clock period" (i.e., time between pulse slots) this sequence (in one field of view) and its logic negation (in another field of view) were used. Since the clock periods ranged from 3 min to 30 min, overall protocol durations ranged from 19×3 = 57 to 19×30 = 570 min. The estimated values of the information transmission rate are means computed for tracks obtained from 2 fields of view and 4–5 biological replicates for each clock period, except for the clock periods of 20 min and 30 min, for which 10 fields of view from a single biological replicate were used.

In the case of interval encoding, the inter-pulse intervals were chosen in a randomized manner: the interval lengths were first selected to best reflect the underlying distribution, either geometric or geometric with a minimal gap, given the time budget of a single experiment (~26 h), and then randomly shuffled. A four hour-long resting period was added in the middle of the experiment to allow the cells to regenerate in 3 of 6 experiments with stimulation according to the protocol without a gap and 5 of 9 experiments with stimulation according to the protocol with a minimal gap; however, no significant difference was observed between results of the experiments with and without the resting period. The resting period and the initial 90-min starvation period were excluded from the analysis. The first pulse of the sequence and the first pulse after the resting period were also discarded as non-representative. Overall, in each protocol there were 24–131 analyzed pulses (during 25–31 h) depending on the assumed average interval between pulses.

## Nuclei detection and cell tracking

The nuclei were detected in the channel of fluorescently tagged histone 2B (H2B)-miRFP703 using local thresholding. Outlines of overlapping nuclei were split based on geometric convexity defects when possible. Outlines of nuclei that were partially out of frame were excluded from analysis. The nuclei were tracked automatically using a greedy algorithm based on parameters such as proximity of outlines in subsequent time points, their surface area, eccentricity, orientation, total fluorescence intensity and intensity distribution. ERK KTR trajectories were obtained by calculating within each tracked nuclear outline the mean intensity in the ERK KTR channel. All image processing was performed within our custom software, SHUTTLETRACKER.

## Signal processing

For each tracked nucleus, the mean ERK KTR intensity in its contour was quantified and normalized, first with the whole-image mean intensity in the ERK KTR channel (to compensate for possible global changes in fluorescence), and then with the average over the cell 120-min history (to account for variation in ERK KTR expression or its uneven visibility in individual cells). Such normalized values were subtracted from 1 and then referred to simply as 'ERK KTR translocation' and denoted $x_t$, where $t$ indicates a time point and directly corresponds to the minute of the imaged part of the experiment. Of note, for nuclear ERK KTR signal constant in time, such normalization and linear transformation would imply $x_t = 0$.

## Selection of tracks

In each field of view, we ranked tracks by their length and then by their quality, defined as low variation of the nuclear area within the track, and preselected 500 top scoring tracks. To eliminate cells that responded weakly to stimulation, for example due to low reporter expression, we further rejected 100 tracks (20%) with the lowest sum-of-squares of the discrete derivative of the ERK KTR translocation trajectory (computed after only the first step of normalization).

## Input reconstruction

**Binary encoding.** For each experiment, 200 out of the 400 ERK KTR translocation trajectories were randomly selected as a train set, and 200 trajectories from cells stimulated with the negated sequence as the test set. For each potential light pulse time point $t_0$, referred to as "slot", we extracted slices of $\ell = 6$ subsequent time points beginning at three different shifts with respect to $t_0$: 2 min before the slot (that is, from $t_0 - 2$ to $t_0 + 3$), 1 min before the slot (from $t_0 - 1$ to $t_0 + 4$), and exactly on the time point of the slot (from $t_0$ to $t_0 + 5$). For each slice we computed $\ell - 1$ discrete backward derivatives, $\Delta x_t = x_t - x_{t-1}$. Separately for each of the three sets of slices (corresponding to a specific shift with respect to $t_0$), a $k$NN ($k = 20$) classifier [43] with ($\ell - 1$)-dimensional Euclidean distance was trained and evaluated so that each slice in the test set was labeled with 0/1 depending on the predicted occurrence of a light pulse in its slot. In this way, three 0/1 labels were assigned to a single slot. To obtain input signal reconstruction in each slot of $x_t$, an ensemble classifier combined three binary predictions through hard voting, *i.e.*, pulse is confirmed if two or three predictions are 'yes'. In the estimation of the confusion matrix (described further), the whole procedure was repeated for 10 random resamplings of the training set and the test set. We checked that the reconstruction is not improved for higher $k$.

**Interval encoding.** The ERK KTR translocation trajectories from each experiment were split into the training set (200 randomly selected trajectories) and the test set (200 remaining trajectories). Each trajectory was represented as a set of overlapping slices of $\ell = 6$ subsequent time points. The slices constitute a perfect 6-fold coverage meaning that each (non-terminal) time point belongs to 6 (partially overlapping) slices. Each slice was labeled with the "time after pulse" (TAP) being the time from the last light pulse to the last time point of the slice (or from the previous pulse if this time is shorter than 3 min); this adjustment is important in the case of very short, 3–4 min, inter-pulse intervals. For each slice we computed $\ell - 1$ discrete backward derivatives, $\Delta x_t = x_t - x_{t-1}$. A $k$NN ($k = 20$) classifier [43] with ($\ell - 1$)-dimensional Euclidean distance was trained to predict a TAP associated with each slice. Slices with TAP values of 3, 4, or 5 min were used to predict pulses through hard voting: time points indicated by at least two of three slices were considered as time points having a light pulse in the final reconstruction (see S2 Fig, panels A and B). To additionally prevent a single pulse from being predicted multiple times, if more than one prediction was indicated within any three subsequent time points, all predictions except the first one were discarded. In the estimation of the contingency table (described further), the whole procedure was repeated for 10 different random partitionings of trajectories into the test set and the train set.

Let us notice that in both the binary and the interval encoding protocols, the decision whether to classify a given time point $t_0$ as a point containing a light pulse was made based on 3 subsequent, partially overlapping slices of length 6, which cover 8 subsequent values of $x_t$ (for $t$ ranging from $t_0 - 2$ to $t_0 + 5$). Thus, only information available to a cell 5 min after the pulse was used by the trained ensemble classifier to make a prediction. In Fig 3D and 3E, the length of this window is varied to check how fast information is accumulated.

## Bitrate computation

We estimated the bitrate as the amount of information $I(S, R)$ between the input signal $S$ and the elicited response $R$ sent within an interval corresponding to the total input duration $\Delta t$ according to the formula:

$$i(S, R) = \frac{I(S, R)}{\Delta t} = \frac{H(S) - H(S|R)}{\Delta t} = \frac{H(S)}{\Delta t} - \frac{H(S|R)}{\Delta t} \tag{3}$$

where $H(S)$ is the entropy of an input signal and $H(S|R)$ is the entropy of the input conditioned on the response in the case of reconstruction-free approach or on the output-based reconstruction of the input in the case of reconstruction-based approach. As described below, the input entropy rate, $h(S) \coloneqq H(S)/\Delta t$, may be determined theoretically, whereas the conditional entropy $H(S|R)$ is calculated based on ERK KTR translocation trajectories.

## Input entropy rate

In the case of the binary encoding protocol, input sequences contain 0/1 digits occurring independently with identical probabilities of ½ in temporally equidistant pulse slots. The entropy of such 19-digit input sequences is $H_{\text{binary}}(S) = 19$ bits. When digits occur in pulse slots $\tau_{\text{clock}}$ apart, then the input entropy rate per digit is

$$h_{\text{binary}}(S; \tau_{\text{clock}}) = \frac{19 \text{ bits}}{19\,\tau_{\text{clock}}} = 1 \text{ bit}/\tau_{\text{clock}} \tag{4}$$

Since $\tau_{\text{clock}}$ is the clock period of the binary encoding protocol, Eq (4) expresses the input entropy rate per clock period.

In the case of the interval encoding protocols, intervals between pulses are drawn at random from the geometric distribution with the rate parameter $p = 1/\tau_{\text{geom}}$ and are optionally lengthened by adding $\tau_{\text{gap}}$. The entropy of the geometric distribution is

$$-\frac{p \log_2 p + (1-p) \log_2 (1-p)}{p} = \tau_{\text{geom}} \log_2 \tau_{\text{geom}} - \left(\tau_{\text{geom}} - 1\right) \log_2 \left(\tau_{\text{geom}} - 1\right) \tag{5}$$

so the input entropy rate $h_{\text{interval}}(S; \tau_{\text{geom}}, \tau_{\text{gap}})$ is

$$h_{\text{interval}}\left(S, \tau_{\text{geom}}, \tau_{\text{gap}}\right) = \frac{\tau_{\text{geom}} \log_2 \tau_{\text{geom}} - \left(\tau_{\text{geom}} - 1\right) \log_2 \left(\tau_{\text{geom}} - 1\right)}{\tau_{\text{geom}} + \tau_{\text{gap}}} \tag{6}$$

where $\tau_{\text{geom}} + \tau_{\text{gap}}$ is the mean inter-pulse interval. Since in the interval encoding protocols, digits 0/1 occur at the one-minute resolution, $h_{\text{interval}}$ computed in Eq (6) is also the entropy per digit. For the interval encoding protocol (nominally without a minimal gap), $\tau_{\text{geom}}$ was varied and a technical $\tau_{\text{gap}}$ of 2 min was introduced in experiments to avoid ambiguities in signal reconstruction. For the protocol with a minimal gap, $\tau_{\text{gap}}$ was varied and $\tau_{\text{geom}}$ was adjusted through numerical optimization to maximize $h_{\text{interval}}(S; \tau_{\text{geom}}, \tau_{\text{gap}})$.

## Conditional entropy—Reconstruction-based approach

For the binary and the interval encoding protocols, we estimate from above the conditional entropy $H(S|R)$ per digit of the input given its reconstruction (throughout this section, $R$ denotes input reconstruction). In light of Eq (3) this will provide us with a lower bound on the bitrate $i(S, R)$. Entropy of a joint distribution is lower than the sum of individual entropies

[44], thus

$$H(S|R) \leq \sum_d H(S_d|R) \tag{7}$$

where $S_d$ is a single input digit and $H(S_d|R)$ is the entropy of that input digit conditioned on the whole-sequence reconstruction $R$. The equality holds only when $R$ and $S$ jointly are a Markov process, this is, when digits occur independently and the reconstruction of a given digit does not depend on other digits. Of note, in our experiments the input digits are independent for the binary encoding protocol, but dependent for the interval encoding protocol with a (minimal) gap. However, even for the binary encoding protocol, the probability of detecting a pulse (digit '1') depends on the time interval from the previous '1'.

Further, from the data processing inequality [44], for the binary encoding protocol, we have

$$H(S_d|R) \leq H(S_d|R_d) \tag{8}$$

where $H(S_d|R_d)$ is the entropy of an input digit $S_d$ conditioned on its reconstruction $R_d$, and for the interval encoding protocols we have

$$H(S_d|R) \leq H(S_d|(R_{d-1}, R_d, R_{d+1})) \tag{9}$$

where $H(S_d|(R_{d-1}, R_d, R_{d+1}))$ is the entropy of $S_d$ conditioned on three subsequent digits of the reconstruction.

Taken together, from inequalities (7)–(9) we have

$$H(S|R) \leq \sum_d H(S_d|R_d) \text{ and } H(S|R) \leq \sum_d H(S_d|(R_{d-1}, R_d, R_{d+1})) \tag{10}$$

which means that to obtain upper bounds on conditional entropies for the binary encoding and the interval encoding protocols we have to calculate $H(S_d|R_d)$ and $H(S_d|(R_{d-1}, R_d, R_{d+1}))$, respectively.

**Calculation of $H(S_d|R_d)$ for binary encoding.** To obtain $H(S_d|R_d)$, we calculated the confusion matrix between $S_d$ and $R_d$:

$$
\begin{array}{ccc}
 & & R_d \\
 & & 1 \quad 0 \\
 & 1 & TP \quad FN \\
S_d & & \\
 & 0 & FP \quad TN
\end{array}
\tag{11}
$$

where TP, FP, FN, TN are, respectively, the probabilities of true positive detections, false positive detections ("false detections"), false negative detections ("missed pulses"), and true negative detections averaged over all time points, all selected tracks, and all partitions of data into the train set and the test set. Based on this confusion matrix we calculated $H(S_d|R_d)$ according to the definition as:

$$H(S_d|R_d) = -E(\log_2 p(S_d|R_d)) = -\sum_{s,r \in \{0,1\}} p(S_d = s, R_d = r) \log_2 \frac{p(S_d = s, R_d = r)}{p(R_d = r)} \tag{12}$$

**Calculation of $H(S_d|(R_{d-1}, R_d, R_{d+1}))$ for interval encoding.** For this encoding, we computed the contingency table showing the relation between $S_d$ and the reconstruction of the three subsequent digits $(R_{d-1}, R_d, R_{d+1})$, which also accounts for information carried by

inaccurate detections:

$$
\begin{array}{cc}
 & (R_{d-1},\ R_d,\ R_{d+1}) \\
 & \begin{array}{cccc}
(0,0,0) & (1,0,0) & (0,1,0) & (0,0,1)
\end{array} \\
S_d \begin{array}{c} 1 \\ 0 \end{array} &
\begin{array}{cccc}
P_{(0,0,0);1} & P_{(1,0,0);1} & P_{(0,1,0);1} & P_{(0,0,1);1} \\
P_{(0,0,0);0} & P_{(1,0,0);0} & P_{(0,1,0);0} & P_{(0,0,1);0}
\end{array}
\end{array}
\tag{13}
$$

The entries of the contingency table are joint probabilities $p_{(r_-,\ r0,\ r+);s} = p(S_d = s, (R_{d-1}, R_d, R_{d+1}) = (r_-, r_0, r_+))$ of input signal $s \in \{0, 1\}$ and reconstructed input $(r_-, r_0, r_+) \in \{(0,0,0),\ (1,0,0), (0,1,0), (0,0,1)\}$. In contrast to the confusion matrix for the binary encoding, these joint probabilities have no straightforward interpretation in terms of TP, FP, FN, TN or inaccurate detection probabilities. The combinations of $(R_{d-1}, R_d, R_{d+1})$ containing two or three 1s do not occur due to the prior elimination of detections that are closer than 3 min apart. The conditional entropy was calculated according to the definition as:

$$
H\big(S_d | (R_{d-1}, R_d, R_{d+1})\big) = -\sum_{s,r_-,r_0,r_+ \in \{0,1\}} P_{(r_-,r_0,r_+);s} \log_2 \frac{P_{(r_-,r_0,r_+);s}}{P_{(r_-,r_0,r_+)}}
\tag{14}
$$

To summarize, bitrate may be computed based on Eq (3) with input entropy rates given by Eqs (4) and (6) and conditional entropies given by Eqs (12) and (14) in the case of the binary encoding protocols and in the case of interval encoding protocols, respectively.

## Reconstruction-based approach—Sources of information loss

To determine the sources of the information loss (Fig 2B, 2D and 2F), we sequentially corrected all types of errors in the reconstruction: false detections (false positives), missed pulses (false negatives), and for the interval encoding protocols also inaccurate (deferred or advanced) detections. After each correction step, we recomputed the confusion matrix/contingency table and attributed the decrease of conditional entropy and the resulting increases of bitrate to the particular type of error. Since the obtained results depend on a particular order of correction steps, to compare individual contributions from the three corresponding types of reconstruction errors, we calculated each of these contributions by averaging over all possible orders of the correction steps (2 permutations for the binary encoding protocol; 6 permutations for the interval encoding protocols; see S2 Fig, panels C–E). Of note, each sequence of correction steps restores exactly the input signal and the estimated bitrate losses associated with considered error types sum up to the total bitrate loss. Since the procedure is symmetric with respect to each type of error, these contributions may be compared.

## Conditional entropy—Reconstruction-free approach

To estimate the information transmission rate directly, *i.e.*, without the reconstruction step (Fig 4), we split the trajectories into slices of length $\ell = 8$ as described above and labeled them with 0/1 depending on whether an input pulse occurred at the time point 5 min before the last time point in the slice. The slice length $\ell = 8$ was chosen to comply with the reconstruction-based approach, where we used three consecutive slices of length 6 that jointly span over an interval of 8 time points. Seven ($= \ell-1$) finite differences were computed for each slice, $r$, embedding all slices in the 7-dimensional Euclidian space. For each $r$, $k = 20$ nearest neighbors were found (including the slice itself) and the numbers $n_0$ and $n_1$ of neighbors marked with 0 and 1, respectively, were counted ($n_0 + n_1 = k$). For each slice, the conditional entropy was

estimated according to the formula:

$$H(S|r) = -\left(\frac{n_0}{k} \log_2 \frac{n_0}{k} + \frac{n_1}{k} \log_2 \frac{n_1}{k}\right) + \text{bias correction} \qquad (15)$$

where (Miller–Madow) bias correction is $1/(2k \ln 2)$ if both $n_0$ and $n_1$ are non-zero, and 0 otherwise [45,46]. The overall conditional entropy $H(S|R)$ was estimated by averaging $H(S|r)$ over all slices $r$. The procedure was performed on 200 trajectories randomly chosen out of 400 cell trajectories in each experiment, and repeated 10 times for different choices to evaluate the robustness of the estimate.

## Supporting information

**S1 Fig. Experimental and data analysis workflow (reconstruction-based approach).**
(TIF)

**S2 Fig. Interval encoding: Input sequence reconstruction algorithm and determination of the sources of information loss (reconstruction-based approach).** (**A**) Tracks from the training set are segmented into overlapping slices of 6 consecutive time points such that each time point belongs to 6 slices. Each slice is labeled with the time after pulse (TAP), measured with respect to the last time point in the slice. The $k$NN classifier is trained to predict the TAP based on 5 discrete differences between 6 consecutive time points of a slice. (**B**) For each slice, the TAP predicted by the classifier indicates a time point at which the stimulation pulse could occur. Votes for particular time points from different slices are counted. Only votes from slices predicted with TAP = 3,4,5 min are taken into account, the remaining are ignored as unreliable. Time points that received at least two out of the three possible votes are considered as time points with pulse in the final reconstruction, $R$. (**C**) Detection and labeling of errors in the reconstruction $R$. Inaccurate detections (one minute before or after the pulse) are not decomposed into missed pulses and false detections but assigned to their specific error types. Since the patterns '11' and '101' are guaranteed never to occur in the input sequence $S$, the error classification is unambiguous. (**D**) An example 3-step sequence of error corrections for the interval encoding protocols. After the third step, the fully corrected reconstruction $R'''$ is identical to the input sequence $S$. The difference between bitrate before and after each correction step is attributed to the particular information loss source. (**E**) As the difference in bitrate depends on the sequence of corrections, the contributions of the three types of errors are averaged over all permutations of correction steps.
(TIF)

**S3 Fig. Cell-to-cell bitrate variability and exclusion of non-responding cells (reconstruction-based approach).** (**A**) Bitrate estimate as a function of the fraction of cells excluded in the preselection step. The (lower bound for) bitrate is computed as in Fig 3. Error bars denote standard error of the mean based on the 2–4 experiments with highest bitrate. In the preselection step we aim at excluding cells that do not respond to stimulation, possibly due to low expression of optoFGFR or ERK KTR. The rejection criterion is formulated in a way *a priori* independent of the accuracy of pulse detection, see Methods for details. Throughout the paper, the fraction of rejected cells is set to 20% (highlighted in gray), because above this value the bitrate estimates in the interval encoding protocols (with and without minimal gap) reach a plateau. (**B, C, D**) Histograms of the information transmission rates in single cells for (B) binary encoding, (C) interval encoding, and (D) interval encoding with a minimal gap. Estimates for cells rejected in the preselection step (20% of all cells) are marked in orange. Negative

bitrate estimates can occur due to the rough approximation based on inequalities in Eq (7) in the main text.
(TIF)

## Acknowledgments

We thank Prof. Joan S. Brugge from Harvard Medical School, Boston, MA for providing the MCF-10A human mammary epithelial cells line. We acknowledge support of the Microscopy Imaging Center of the University of Bern (https://www.mic.unibe.ch).

## Author Contributions

**Conceptualization:** Paweł Nałęcz-Jawecki, Paolo Armando Gagliardi, Marek Kochańczyk, Olivier Pertz, Tomasz Lipniacki.

**Data curation:** Marek Kochańczyk.

**Funding acquisition:** Olivier Pertz, Tomasz Lipniacki.

**Investigation:** Paweł Nałęcz-Jawecki, Paolo Armando Gagliardi.

**Methodology:** Marek Kochańczyk, Coralie Dessauges.

**Resources:** Paolo Armando Gagliardi, Coralie Dessauges.

**Software:** Marek Kochańczyk.

**Visualization:** Paweł Nałęcz-Jawecki, Marek Kochańczyk.

**Writing – original draft:** Paweł Nałęcz-Jawecki, Marek Kochańczyk, Tomasz Lipniacki.

**Writing – review & editing:** Paolo Armando Gagliardi, Olivier Pertz, Tomasz Lipniacki.

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
