## [Decision Letter · Decision Letter 0]

18 Feb 2023

Dear Prof. Lipniacki,

Thank you very much for submitting your manuscript "The MAPK/ERK channel capacity exceeds 6 bit/hour" for consideration at PLOS Computational Biology.

As with all papers reviewed by the journal, your manuscript was reviewed by members of the editorial board and by several independent reviewers. In light of the reviews (below this email), we would like to invite the resubmission of a significantly-revised version that takes into account the reviewers' comments.

We cannot make any decision about publication until we have seen the revised manuscript and your response to the reviewers' comments. Your revised manuscript is also likely to be sent to reviewers for further evaluation.

Sincerely,

Padmini Rangamani

Academic Editor

PLOS Computational Biology

Kiran Patil

Section Editor

PLOS Computational Biology

Reviewer's Responses to Questions

**Comments to the Authors:**

Reviewer #1: This study focuses on the information transmission of ERK pathway in face of a sequence of pulses. The authors used different sequences of light pulses to activate the optoFGFR, and then measured the ERK activity through the fluorescent ERK KTR. Based on the time courses of ERK KTR translocation, authors calculated the transmitted information rate using classifier-based estimation and obtained the value of transmitted information 6 bit/hour. The experiment design and information estimation are air-tight. Aside from the tools, the main conclusion is, to me, very impressive. The 6 bit/hour is different from the transmitted information when the external stimulus is just varied once, which is usually near 1 bit. The conclusion provides insight into how the ERK pathway transfers information.

While this is a thoughtful study, it can still be improved by addressing the accuracy and robustness of main conclusion. My comments are as follows:

Major comments

1. The authors used the output-based reconstruction to estimate the conditional entropy. However, the reconstruction seems not necessary for the estimation, because the conditional entropy can be directly estimated from the original output by using other methods, such as k-nearest neighbor (kNN) based estimators and kernel-based methods. Thus, justifications are needed to explain why the reconstruction is required for the conditional entropy estimation. Furthermore, it’s unclear whether the reconstruction-based method provides an accurate estimation of conditional entropy. Therefore, a direct comparison with other widely used methods will provide a much more vigorous justification.

2. It is unclear how robust the main conclusion is. Therefore, error bars showing the variance of mutual information are needed, for example, in Fig 2B, 2D, 2F, and Fig 3. This will provide the robustness of the estimated mutual information value 6 bit/hour.

3. The authors need to go further to convey the motivation for choice of protocols. It would be helpful to describe what type of stimulus the cell usually faces, a constant stimulus or a serial of random pulse? What is the reason to consider binary encoding, interval encoding, and interval encoding with minimal gap? The authors need to justify the biological implications and choice of the three types of pulsatile stimulation protocols.

4. It is confusing about the difference between the three protocols mentioned in Fig. 1A. The study uses geometry distribution in the second and third protocols. Since the geometry distribution with mean tau_geom is the probability distribution of the number X of Bernoulli trials needed to get one success, it means that a binomial trial is made at every minute and the probability of success is 1/tau_geom. Besides, the first protocol is that a binomial trial is made every 5 minutes and the probability of success is 1/2. So, if I understand correctly, the second protocol is very similar to the first protocol if tau is 2, and the third protocol is the same as the first protocol if tau_geom is 2 and tau_gap is 5.

5. The work performs experiments with the plus duration 100ms. Different plus duration may encode different mutual information. Besides, the study used one-minute resolution. Could the authors discuss more about the possible effects when changing plus duration and one-minute resolution?

6. In the Discussion, the author discussed the information transmission rate in other biological systems, for example, neurons, E. coli chemotactic system. Could the authors compare the information transmission rate between ERK pathway and these systems and then discuss the possible reason to cause such difference?

Minor comments

1. The description of the interval encoding protocol with a minimal gap in lines 82-85, “…the time intervals were also drawn from a geometric distribution (with the mean interpulse interval tau_geom), but the intervals shorter than a minimal interval tau_gap were excluded…”, could be misleading. According to such a description, it’s unclear how the expectation value of time interval equals tau_geom + tau_gap, as shown in Fig. 1A and in the method section.

2. The “gapless protocol” in lines 450-451 is confusing. Which one of the three types of pulsatile stimulation protocols does it indicate?

3. On Page 22, “For the "gapless" protocol, tau_geom was varied whereas tau_gap was set to 2 min (to avoid ambiguities in signal reconstruction). For tau_geom > 3.15 min, hinterval(S; tau_geom, tau_gap) is a decreasing function of tau_geom…”. The paragraph is hard to understand and needs to be rewritten. For example, it’s confusing that a “gapless” protocol has a tau_gap = 2 min; it is unclear what is the purpose of varying tau_geom; how is the “3.15 min” determined?

4. In Fig. 1A, the way to plot time duration 5,6,21,3,18,… can be misleading. It would be more clear if a curly bracket is placed above these time durations or if these numbers are rewritten in a monotonically increasing way, for example 0, 5 min, 10 min, 15 min, 20 min, …

5. The legend for small gray bars in Fig. 1D is missing.

6. The y axis label “Information transmission rate” in Fig. 2 B, D, and F is confusing. The information transmission rate is the transmitted rate, that is, the black part of a bar, not the whole bar.

Reviewer #2: In this paper, the authors measure a rate of information transfer through the Ras-ERK signaling pathway in fibroblast cells. This was done using an optogenetic stimulation system and ERK nuclear translocation reporter for measuring cell responses. The authors delivered three stimulus protocols in which signal variations were encoded differently (presence or absence of signal at each time; timing of signals; timing of signals with a minimum duration between consecutive signals). The mutual information between the stimulus trajectory and cell response trajectory was estimated using a classifier that was trained to detect when signals were delivered, and a mutual information rate was calculated by dividing by the total trajectory time. The authors estimate an information rate of 4-6 bits/hr under these protocols.

My main concern is about the motivation and interpretation of these results. What is the biological meaning of this information rate? What makes it "the" information rate (line 271)?

The cell responses and thus the information rates likely depend on strength of optical illumination. How did the authors choose the illumination strength? How do the measured responses with this illumination strength compare to responses to physiological levels of chemical inputs? And thus, how does the measured information rate compare to what might be expected in a biological context?

It is also hard for me to assess whether these protocols are close to anything cells would see in natural contexts, and thus to assess whether the measured information rates mean anything to the cell.

Several scenarios are mentioned in the Discussion where Ras-ERK signaling is used in higher organisms, but not all of those involve ERK nuclear translocation. It seems like the measured information rate is irrelevant to those processes.

Maybe this work is relevant to synthetic biology? One might want to use the Ras-ERK pathway to send signals through a human-made biological device?

Other:

This paper and others before it lump together responses of different cells. But this is known to reduce the measured information transfer because cell-to-cell differences in responses are treated as noise, even if single-cell responses are very reproducible (e.g. as in the data of Toettcher et al, ref 22). The authors should comment on the difference between single cell and population-lumped measurements and what it means for how their results are interpreted.

Related, is there communication among cells in this experiment, as it sounds like around line 289 in the Discussion?

How was the 8-minute window size chosen for the classifier? Does this have biological significance?

In the introduction and conclusion, the authors estimate an information rate of 1 bit per minute during Dicty chemotaxis. This estimation procedure only works if the cells always switch direction with the chemotactic gradient when it oscillates at 0.02 Hz. If there is any variability (incorrect decisions), the information rate must be lower.

**Have the authors made all data and (if applicable) computational code underlying the findings in their manuscript fully available?**

Reviewer #1: None

Reviewer #2: Yes

PLOS authors have the option to publish the peer review history of their article (what does this mean?). If published, this will include your full peer review and any attached files.

Reviewer #1: No

Reviewer #2: No
---

## [Decision Letter · Decision Letter 1]

4 May 2023

Dear Prof. Lipniacki,

We are pleased to inform you that your manuscript 'The MAPK/ERK channel capacity exceeds 6 bit/hour' has been provisionally accepted for publication in PLOS Computational Biology.

Best regards,

Padmini Rangamani

Academic Editor

PLOS Computational Biology

Kiran Patil

Section Editor

PLOS Computational Biology

Reviewer's Responses to Questions

**Comments to the Authors:**

Reviewer #1: I thank the authors for carefully addressing all the issues. There are no remaining issues. A minor suggestion is to make the x label in Fig 3B (tau_geom+2) consistent with that in Fig 2D (tau_geom).

Reviewer #2: The authors have mostly addressed my concerns.

The only thing is that I did not find text added to the manuscript about how they chose their light stimulation intensity. The first argument in the response to my comment, that lower light intensity causes some cells to not respond, does not seem relevant--this is valid information loss. Their second argument that the light intensities used elicit ERK responses that resemble responses to physiological levels of growth factors is more convincing and should be included somewhere in the text.

**Have the authors made all data and (if applicable) computational code underlying the findings in their manuscript fully available?**

Reviewer #1: Yes

Reviewer #2: Yes

PLOS authors have the option to publish the peer review history of their article (what does this mean?). If published, this will include your full peer review and any attached files.

Reviewer #1: No

Reviewer #2: No

---

## [Editor Report · Acceptance letter]

16 May 2023

PCOMPBIOL-D-23-00079R1 

The MAPK/ERK channel capacity exceeds 6 bit/hour

Dear Dr Lipniacki,

I am pleased to inform you that your manuscript has been formally accepted for publication in PLOS Computational Biology. Your manuscript is now with our production department and you will be notified of the publication date in due course.

With kind regards,

Zsofia Freund
